# ESMfluc: Predicting Flexible Regions in a Protein Using Language Models

## Abstract

Proteins are dynamic molecular machines whose functionality emerges not merely from their static structures but critically from their intrinsic conformational flexibility. Understanding how a protein sequence encodes this flexibility is essential for deciphering the connection between sequence, dynamics, and biological function. While recent advances in deep learning and protein language models have significantly improved structural prediction, predicting sequence-encoded dynamics remains challenging. In this work, we introduce ESMfluc, a biLSTM model trained on molecular dynamics simulation data, utilizing embeddings from the Evolutionary Scale Modeling (ESM) architecture to predict local flexibility directly from protein sequences. Using fluctuation data derived from extensive molecular dynamics simulations, ESMfluc accurately identifies flexible residues without computationally expensive simulations while providing interpretability via attention maps. The model notably highlights distal flexible regions relevant for allosteric regulation and drug targeting. Our approach demonstrates substantial improvements over traditional flexibility proxies, offering researchers a computationally efficient method to reveal critical functional sites beyond active or binding regions.

## 1 Introduction

Proteins are dynamic entities whose function critically depends on conformational changes and motion Henzler-Wildman & Kern (2007); Koshland Jr (1958). Dynamic fluctuations enable processes such as allosteric regulation, molecular recognition, and enzymatic catalysis, which cannot be explained by static structures alone Frauenfelder et al. (1991); Alavi et al. (2024). Indeed, the importance of protein dynamics extends to practical applications; for example, in drug design a protein's flexibility can create transient cryptic binding pockets and influence ligand binding, meaning that accounting for target dynamics is essential for "hitting a moving target" in structure-based drug discovery Carlson (2002); Wei & McCammon (2024); Durrant & McCammon (2011).

Traditional experimental techniques for protein flexibility exist—X-ray crystallographic B-factors report atomic displacement Sun et al. (2019), and NMR spectroscopy can measure backbone mobility or provide ensembles of conformations Kleckner & Foster (2011)—but these approaches are labor-intensive and limited in the timescales or conditions they can probe Carugo (2022); Li & Kang (2017).

Molecular dynamics (MD) simulations offer an alternative means to capture protein motions in silico. However, MD is computationally expensive and, until recently, large-scale standardized databases of protein dynamics were unavailable, making it challenging to train learning-based models on simulation-derived flexibility data. The newly introduced ATLAS database addresses this gap by curating all-atom MD trajectories for hundreds of proteins, providing a broad benchmark set of residue-level flexibility measures derived from simulations Vander Meersche et al. (2024). Such resources pave the way for data-driven models of protein dynamics.

Parallel to these developments in characterizing protein motion, the past few years have seen breakthroughs in static protein structure prediction as artificial intelligence (AI) has solved the decades-old problem of how a protein's sequence determines its structure. Last year, the Nobel Prize in Chemistry was awarded to David Baker for computational protein design, as well as to Demis Hassabis and John Jumper, authors of the AlphaFold model for predicting protein structure Jumper et al. (2021);

Varadi et al. (2022). However, a critical gap still exists- we lack a clear picture of the underlying physics governing the dynamics of large conformational changes which are crucial for sustaining life. Large-scale implementations of AlphaFold2 now provide high-confidence static models for millions of proteins. Yet, AlphaFold's predictions are rigid structures – essentially snapshots of the lowest-energy conformation – and thus do not explicitly capture the flexibility or alternative states that may be functionally relevant. The success of AlphaFold2 highlights that a protein sequence contains rich information about structure; the question remains though whether the sequence can also reveal a protein's propensity for dynamics and disorder.

In this work, we introduce ESMFLUC, an ESM-powered biLSTM-attention network for predicting residue-level flexibility from the amino acid sequence. ESMFLUC builds on the pre-trained ESM-2 model by training on residue-level flexibility metrics derived from MD simulations (in particular, the ATLAS dataset's entropy-based fluctuations per residue). By leveraging the rich contextual representations of a state-of-the-art language model, ESMFLUC can learn the subtle sequence patterns associated with backbone mobility, such as flexible loop motifs or hinge regions, which might be missed by simpler descriptors. Our results show that ESMFLUC accurately identifies flexible regions without requiring any structural input, outperforming baseline predictors that do not use MD-driven data. Recent diffusion-based generators such as BioEmu Lewis et al. (2025) excel at reconstructing global equilibrium ensembles, yet they come with substantial computational and data requirements. In contrast, ESMFLUC focuses on mapping sequence to per-residue local flexibility, runs orders of magnitude faster, and requires no structural templates or MD pre-computation. Therefore, ESM-FLUC serves as a lightweight front-end that can identify hinge residues. In prospective pipelines, ESMFLUC can guide where BioEmu's costly ensemble generation should be focused, rendering the two methods complementary rather than competing.

In summary, predicting protein dynamics from sequence is important for understanding function and designing therapeutics, and deep learning models provide a powerful tool for this challenge. ESMFLUC is, to our knowledge, one of the first sequence-based approaches trained directly on MD-derived residue-level flexibility. By bridging the gap between sequence-based structure prediction and dynamics, our approach offers a novel route to integrating protein motions into genomics-scale analyses, helping to illuminate how proteins breathe, bend, and adapt, as governed by their sequences.

## 2 RELATED WORK

### 2.1 PROTEIN STRUCTURE PREDICTION AND REPRESENTATION LEARNING

If a deep learning model can encode structure, it is plausible that it also encodes signals of local flexibility or disorder, since these properties are intertwined with sequence and structure evolution. Indeed, some studies have found that regions of high uncertainty or low confidence (such as AlphaFold's pLDDT or language-model perplexity) often correspond to flexible or disordered segments Alderson et al. (2023). Additionally, recent advances in protein language models (i.e., deep learning models that learn semantic representations of amino acid sequences by treating proteins analogously to natural language, where amino acids are treated like words) suggest that sequence-based models can learn complex structure-function relationships. To facilitate progress in research on protein embeddings, the Tasks Assessing Protein Embeddings (TAPE) were introduced in 2019 Rao et al. (2019), a set of five biologically relevant semi-supervised learning tasks spread across different domains of protein biology. ProtBERT, a variant of BERT designed for protein sequences, was used to generate protein embeddings Elnaggar et al. (2021).

More recently, the ESM family of Transformer-based language models (Evolutionary Scale Modeling) demonstrated that by training on hundreds of millions of sequences and implicitly learning aspects of protein structure and function Lin et al. (2023). This ability of large language models to generate accurate structures at scale—e.g., predicting structures for hundreds of millions of metagenomic proteins in the ESM Metagenomic Atlas Lin et al. (2023)—indicates that the latent representations of such models capture fundamental biophysical constraints. This suggests that large pre-trained models are a promising foundation for predicting protein dynamics.

In a recent study, a tool named NetSurfP.3 was developed to predict protein structural disorder using embeddings from ESM Høie et al. (2022). Another example is SPOT-Disorder2 which used an

ensemble of deep learning models (bidirectional LSTMs and convolutional networks) to predict per-residue intrinsic disorder Hanson et al. (2019). Such computational predictors of protein disorder from sequence provide a baseline to improve upon. These advances highlight that protein deep learning models can serve as versatile feature extractors, transferring effectively to downstream tasks in protein biology. However, to date, most efforts have focused on predicting static or binary features, such as structure and disorder, rather than continuous dynamics. No model has yet been specifically fine-tuned to predict local flexibility values across the protein structure from the sequence alone.

## 2.2 MOLECULAR DYNAMICS SIMULATIONS

Molecular Dynamics (MD) simulations have long been used to capture the temporal behavior of biomolecules Karplus & McCammon (2002). Classical force-field-based simulations, such as those implemented in GROMACS Abraham et al. (2015) and AMBER Case et al. (2005), make it possible to study millisecond-scale processes such as protein folding. While MD is powerful in principle, the method is computationally expensive and often limited to nanosecond-to-microsecond timescales, far shorter than biologically relevant processes.

## 2.3 MACHINE LEARNING FOR PREDICTING PROTEIN DYNAMICS

Recent work has explored combining MD with machine learning. For example, Reweighted Autoencoded Variational Bayes for Enhanced sampling (RAVE) Lamim Ribeiro & Tiwary (2018) investigated efficient ligand-protein unbinding. Tribello and Gasparotto used dimensionality reduction techniques to analyze protein trajectories Tribello & Gasparotto (2019). More recently, graph neural networks have been proposed to model residue-level interactions and dynamics Jing & Xu (2021), and the Transformer architecture has been applied to MD trajectory data for coarse-grained conformational modeling Mahmoud et al. (2022). Transformer-based variational autoencoders have also been used for generating novel proteins Sevgen et al. (2023). While these studies illustrate that deep learning can uncover meaningful dynamics, they typically rely on training directly on MD simulations data rather than using pretrained protein embeddings, as in our work.

# 3 METHODS

## 3.1 SIMULATION-DERIVED DATASET

We train and evaluate on ATLAS; the first openly curated collection of standardized, fully solvated, all-atom MD trajectories for a representative slice of the Protein Data Bank (PDB). It contains 1,390 unique proteins, selected for crystallographic resolution $\leq 2.5\,\text{Å}$, absence of missing loops, no cofactors/ligands, and $< 40\%$ pairwise sequence identity to minimize redundancy. To create the dataset, they placed each protein in a periodic triclinic box, solvated using TIP3P water molecules, and neutralized with Na+/Cl- ions at a concentration of $150mM$. Then energy-minimization was done using the steepest descent algorithm for $5,000$ steps, followed by a two-step equilibration ($200ps$ NVT where Number of particles, Volume, and Temperature are kept constant, and $1ns$ NPT where Number of particles, Pressure, and Temperature are kept constant). The final production molecular dynamics simulations were carried out in three replicates, $100ns$ each with a time step of $2fs$, using a different seed for the random starting velocities assigned from a Boltzmann distribution. They then assessed the obtained MD trajectories and reported multiple parameters regarding the overall behavior of the protein, including an entropy-based measure of flexibility which is defined per residue as:

$$N_{eq} = 2^H \tag{1}$$

where

$$H = -\sum_{i=1}^{16} p_i log_2(p_i) \tag{2}$$

and $p_i$ is the probability of the residue visiting protein block $i$ during the simulation time. Protein-blocks (PBs) are structural prototypes; in principle, any conformation of any amino acid could be

represented by one of the 16 available PBs de Brevern et al. (2000); Barnoud et al. (2017). A $N_{eq}$ of 1.0 means the amino acid is very rigid, and a $N_{eq}$ of 16 means it visits all possible PBs equally frequently and thus is highly flexible. We chose $N_{eq}$ as the target for our classification task. Fig. 1 shows the distribution of $N_{eq}$ values. in the training set.

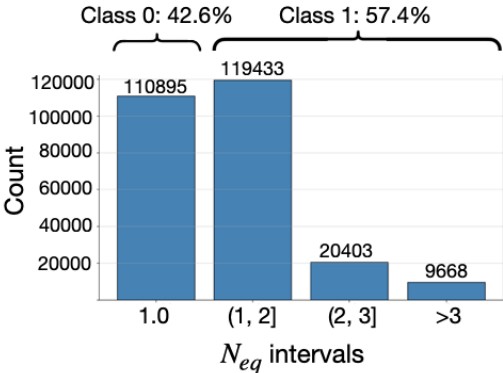

Figure 1: Histogram of $N_{eq}$ values in the training set.

## 3.2 DATA PROCESSING

The total dataset was split into 80% for training and 20% as a held-out test set to ensure a reliable and unbiased assessment of the model's generalization ability. This resulted in 256,304 residues for training and validation and 62,895 residues held out for testing. To prepare the protein sequences for model input, each sequence was first tokenized using the ESM tokenizer. Due to the 1024-token input length limitation of ESM models, sequences exceeding this threshold cannot be processed directly. As a result, longer sequences were excluded from the dataset for this study to ensure compatibility with the model's input constraints. From the ESM-2 (t33/650M) backbone we extracted residue-wise embeddings by taking the last hidden state of the final Transformer layer. For a sequence of length $L$ this yields an $L \times 1280$ matrix (one 1280-dimensional vector per residue token).

## 3.3 LABEL PREPARATION

Fig. 1 shows the long-tail distribution of $N_{eq}$ values in the training set. For binary classification, amino acids with $N_{eq} = 1.0$ are labeled as class 0, indicating a complete lack of flexibility as these amino acids are stuck in one PB configuration, and amino acids with $N_{eq} > 1$ are classified as class 1, indicating that these amino acids visit more configurations and thus are flexible.

## 3.4 MODEL ARCHITECTURE

We use the Hugging Face implementation of ESM2 (33 layers, 650M parameters), and extract per-residue hidden states from the final transformer layer (dimension $H{=}1280$) as fixed features. Unless specified otherwise, the ESM backbone is frozen. We study multiple heads on top of ESM embeddings including the following: **ESM + FC (Linear probe)**: A single fully connected layer maps the $H$-dimensional per-residue embedding to logits for two classes. This serves as a strong, capacity-matched baseline. **ESM + BiLSTM + Self-Attention**: A 3-layer bidirectional LSTM Hochreiter & Schmidhuber (1997) (hidden size 512) models local sequence context. A single-head token–token self-attention layer is added after the BiLSTM to produce context-aware token representations, followed by dropout and an FC classifier. This head yields token-level attention maps that we use for qualitative interpretation. Figure 2 depicts the best-performing variant (ESM + BiLSTM + attention).

To address class imbalance, focal loss was implemented which is defined as Lin et al. (2017):

$$\text{FL} = -\frac{1}{|S|} \sum_{i \in S} \left(1 - p_{i,y_i}\right)^{\gamma} \log p_{i,y_i} \tag{3}$$

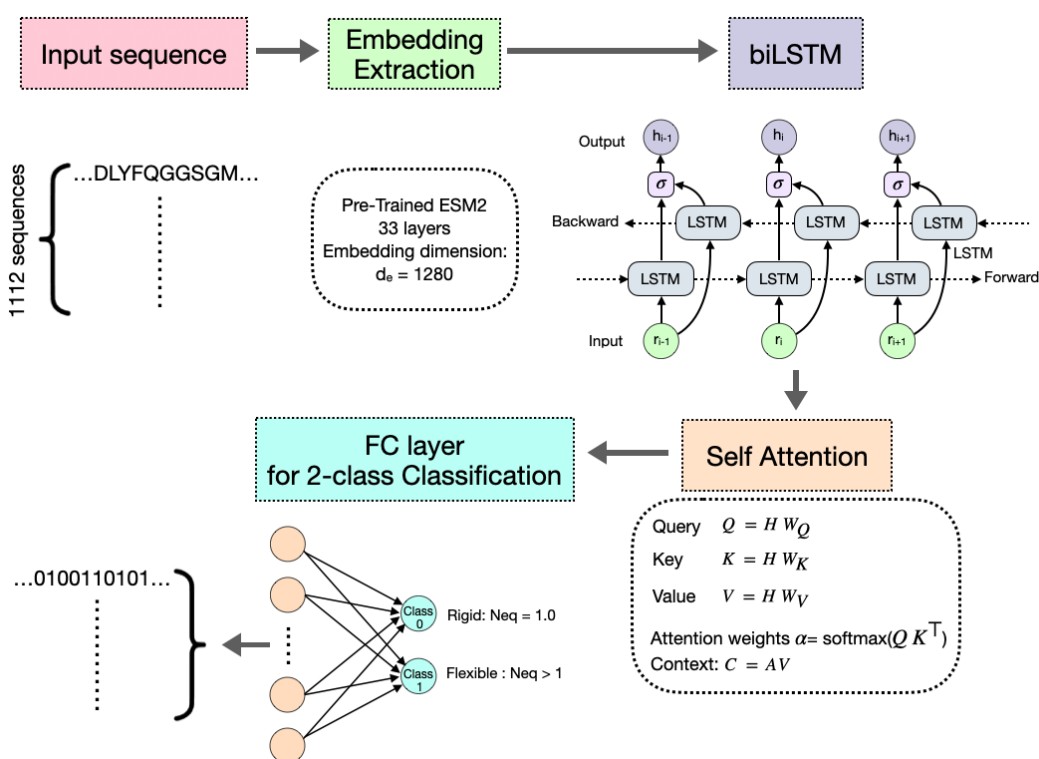

Figure 2: The full pipeline of ESMFLUC model, from ESM embedding extraction to the LSTM with self attention and a final fully-connected layer for binary classification of $N_{eq}$ values.

where $p_{i,y_i}$ denotes the softmax probability of the true class $y_i$ for token $i$ and $\gamma$ is the focusing parameter. Unless noted, we train with batch size 4, weight decay 0.01, dropout 0.3 and focal loss ($\gamma=2$).

## 4    RESULTS

### 4.1    PERFORMANCE OF ESMFLUC

We evaluated the performance of ESMFLUC model using precision, recall, and macro average F1 metrics on the held-out test set. We compared our model to baseline statistical classifiers, ran an ablation study and grid search to tune our model's hyperparameters, explored various loss functions, and investigated how the performance improved as we increased the ESM embeddings' dimensionality as well as batch size.

As shown in Table 1, a linear probe (ESM+FC) already performs strongly, indicating that frozen ESM embeddings encode substantial information about residue dynamics. Sequence models on top of ESM yield consistent gains in sensitivity to flexible residues (Class 1 recall), with BiLSTM+Attention improving the recall–precision balance while providing token-level attention maps for qualitative interpretation. The pure Transformer head underperforms in macro F1 in our setting, trading higher Class 0 precision for lower Class 0 recall, likely reflecting over-confident decisions without the inductive bias of recurrence on frozen features. Overall, the small but repeatable edge of BiLSTM(+Attn) over the linear probe, coupled with interpretability, motivates its use as our primary model.

Table 1: Variants of ESMFLUC versus baselines using ESM2 (33 layers, 650M). Metrics are per-chain and macro-averaged across valid chains. "Transformer" denotes a Transformer head following Vaswani et al. (2017).

| Method | Class 0 | | Class 1 | | Macro F1 | Epochs Ran |
|---|---|---|---|---|---|---|
| | **P** | **R** | **P** | **R** | | |
| FC | 78.6 | 69.0 | 78.8 | 86.0 | 77.9 | 39 |
| LSTM | 78.10 | 73.2 | 80.9 | 84.7 | 79.2 | 18 |
| BiLSTM | 80.8 | 72.3 | 80.8 | 87.2 | 80.0 | 14 |
| LSTM + Attention | 80.0 | 70.6 | 79.9 | 86.9 | 79.1 | 29 |
| BiLSTM + Attention | 82.5 | 69.4 | 79.6 | **89.0** | 79.7 | 18 |
| Transformer | 78.8 | 70.2 | 79.5 | 85.9 | 78.0 | 16 |

## 4.2 CLASSICAL MODEL COMPARISON

For context, we also benchmarked classical scikit-learn baselines trained on the same training split. Logistic Regression (LR) and Random Forest (RF) use frozen ESM2 embeddings with 33 layers and 650 million parameters as features, while Conditional Random Field (CRF) uses the charges, polarity, and hydrophobic properties of amino acids as features. The one-hot Logistic Regression model used a one-hot encoding of each amino acid, previous amino acid's $N_{eq}$ value, the next amino acid's $N_{eq}$ value, and its relative position in the protein sequence. More details on feature engineering and hyperparameters for LR/RF/CRF are provided in Appendix. The large gap between the one-hot and embedding-based LR models demonstrates the importance of using the ESM embeddings. As shown in Table 2, a simple logistic regression is already competitive in macro F1, underscoring how linearly separable the frozen ESM representation is for this task. However, sequence models on top of ESM (Table 1) consistently improve Class 1 recall (flexible residues) and offer token-level interpretability via attention, which the classical baselines lack.

Table 2: Classical baselines trained on the same split. LR/RF use frozen ESM2 features; one-hot LR uses hand-crafted one-hot/positional/neighborhood features; CRF uses physicochemical features. Metrics are per-chain and macro-averaged across chains.

| Method | Class 0 | | Class 1 | | Macro F1 | Time (s) |
|---|---|---|---|---|---|---|
| | **P** | **R** | **P** | **R** | | |
| CRF | 66.2 | 43.9 | 65.8 | 82.8 | 63.1 | 2 |
| LR (One-hot) | 61.1 | 47.1 | 65.5 | 77.1 | 62.0 | 3 |
| RF (One-hot) | 52.5 | 51.7 | 63.4 | 64.1 | 57.9 | 26 |
| LR (ESM) | 78.5 | 72.9 | 80.3 | 84.7 | 79 | 85.42 |
| RF (ESM) | 77.0 | 58.7 | 73.2 | 86.5 | 72.9 | 932 |

## 4.3 SEQUENCE-ONLY VS. STRUCTURE-CONDITIONED: ESMFLUC VS NETSURFP

For our final experiment, we compared ESMFLUC model's predicted probabilities to the predictions from state-of-the-art NetSurfP, which predicts "disorder" from static structures. We evaluate token-level predictions from ESMFLUC against per-residue ground truth dynamics ($N_{eq}$) and a structure-based baseline (NetSurfP disorder). For each test chain, our trained checkpoint (ESM2-650M backbone with an BiLSTM+self-attention head) is applied to the raw amino-acid sequence to produce per-residue logits $\ell_i \in \mathbb{R}^2$ and flexible-class posterior probabilities $p_i = \mathrm{softmax}(\ell_i)_1$. We align these predictions with ground truth $N_{eq}$ values and NetSurfP disorder probabilities. We report two complementary metrics: AUROC for binary discrimination of flexible residues defined by a threshold on $N_{eq}$ ($N_{eq} > 1.0$), and Spearman correlation $\rho$ between the continuous $N_{eq}$ values and each method's continuous score. AUROC captures threshold-free discrimination of flexible vs. rigid residues under a biologically interpretable definition of flexibility (via $N_{eq}$), while Spearman $\rho$ measures monotonic agreement with the magnitude of $N_{eq}$, which is important because dynamics are not purely binary.

Across chains, ESMFLUC substantially outperforms NetSurfP on both discrimination (AUROC) and rank correlation (Spearman). This gap is expected: NetSurfP is trained to infer per-residue

Table 3: A comparison of ESMFLUC model to state-of-the-art NetSurfP in terms of how well their predictions correlate with the ground truth $N_{eq}$ values.

|  | AUROC | Spearman |
|---|---|---|
| **ESMFLUC** | **0.857** | **0.618** |
| NetSurfP | 0.659 | 0.340 |

"disorder" from *static structure features* and thus treats regions that are hard to model structurally as disordered. By contrast, ESMFLUC starts from sequence-only ESM representations and is trained directly on a *dynamics* target ($N_{eq}$-derived labels), enabling it to capture sequence motifs predictive of conformational flexibility beyond what can be inferred from static structure surrogates. The stronger Spearman $\rho$ further indicates that ESMFLUC tracks the *degree* of flexibility: residues with higher $N_{eq}$ tend to receive higher predicted probability, not just cross a binary threshold. Finally, we note that using the posterior $p_i$ consistently matches or exceeds the raw logit as a scoring function, aligning with probabilistic calibration objectives.

## 5 DISCUSSION

### 5.1 INTERPRETABILITY VIA ATTENTION

One advantage of using an attention-based head is that it gives us a window into *how* the model makes predictions. In our self-attention layer, each residue position $i$ is mapped to a query $\mathbf{q}_i$ and each position $j$ to a key $\mathbf{k}_j$. The attention matrix $\boldsymbol{\alpha} \in \mathbb{R}^{L \times L}$ assigns, for every pair $(i,j)$, a weight

$$\alpha_{ij} = \text{softmax}_j\left(\mathbf{q}_i\mathbf{k}_j^\top\right), \qquad \sum_{j=1}^{L} \alpha_{ij} = 1. \tag{4}$$

Intuitively, the dot product $\mathbf{q}_i\mathbf{k}_j^\top$ scores how relevant residue $j$ is for updating residue $i$, and the softmax turns these scores into a probability-like distribution over $j$ for each query $i$. Because $\boldsymbol{\alpha}$ is an *explicit, directed* dependency map, we can use it to infer what the model relies on when predicting flexibility. "Incoming" weights ($\alpha_{\cdot i}$) highlight residues that most influence residue $i$'s prediction (putative regulators or hinges), while "outgoing" weights ($\alpha_{i\cdot}$) indicate residues that position $i$ consults broadly (information hubs). In practice, overlaying high-weight pairs ($i \leftrightarrow j$) on a structure reveals *putative mechanical pathways* (hinges/relays) that underlie the learned flexibility signal. Attention is a powerful hypothesis generator, not a guaranteed causal explanation; combining it with gradient/occlusion analyses or targeted perturbations strengthens mechanistic claims.

Figure 3 shows the attention heat map for PDB: 1LRI, highlighting key residue–residue interactions; Figure 4 shows the corresponding crystal structure. The annotation bars indicate the predicted secondary structure (NetSurfP Høie et al. (2022)) and the predicted $N_{eq}$ class from ESMFLUC. The heat map reveals residues that strongly influence each other, including positions that are far apart in sequence (e.g., residues 38 and 78), consistent with long-range coupling important for flexibility.

### 5.2 QUANTIFYING ATTENTION HOMOPHILY

We measure how much a query residue attends to keys that share its *secondary structure* class. Let $s_j \in \{C, H, E\}$ be the NetSurfP 3-state secondary structure of key $j$, encoded as one-hot vectors:

$$\mathbf{e}_C = (1,0,0), \qquad \mathbf{e}_H = (0,1,0), \qquad \mathbf{e}_E = (0,0,1).$$

For a query $i$, define its attention distribution over secondary structure (SS) classes as

$$\overrightarrow{\mathbf{A}}_i = \sum_{j=1}^{L} \alpha_{ij}\, \mathbf{e}_{s_j} = \left(A_i^C,\, A_i^H,\, A_i^E\right), \qquad \sum_{k \in \{C,H,E\}} A_i^k = 1. \tag{5}$$

Averaging over all queries that themselves belong to class ss $\in \{C, H, E\}$,

$$\overline{\mathbf{a}}^{(\text{ss})} = \frac{1}{N_{\text{ss}}} \sum_{i:\, s_i = \text{ss}} \overrightarrow{\mathbf{A}}_i, \tag{6}$$

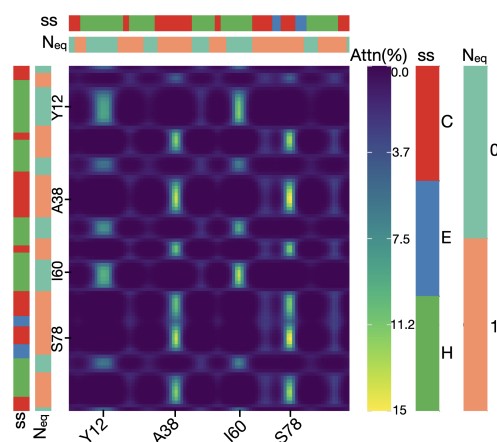

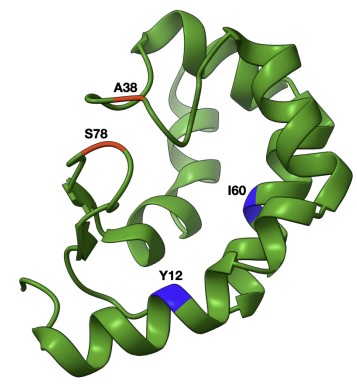

Figure 3: Residue level attention mechanism in a test set sample (PDB 1LRI). Annotation bars: top/left are secondary-structure prediction (red=C,green=H,blue=E) obtained from NetSurfP4. bottom/right are predicted $N_{eq}$ class (orange=1, teal=0)

Figure 4: Crystal Structure of Sterol carrier protein Cryptogein (PDB 1LRI). Residues marked in red are coils with predicted $N_{eq}$ class of 1 (flexible) which act as information hubs and are consulted broadly. Residues marked in blue are helices with predicted $N_{eq}$ class of 0 (rigid) which significantly influence other helices.

yields the typical attention allocation for that class. Over the full test set, coils allocate $\approx 73\%$ of their attention to other coils, helices to helices ($\approx 59\%$), and strands attend more to coils ($\approx 56\%$) than to strands ($\approx 23\%$). This suggests the model learns SS-aware neighborhoods while still consulting flexible coils as information hubs.

We perform an analogous analysis for the predicted flexibility class. Let $n_j \in \{0, 1\}$ be the $N_{eq}$ class (0: rigid, 1: flexible) of key $j$, and encode it as

$$\mathbf{q}_0 = (1, 0), \qquad \mathbf{q}_1 = (0, 1).$$

For query $i$, define

$$\overrightarrow{\mathbf{B}}_i = \sum_{j=1}^{L} \alpha_{ij} \mathbf{q}_{n_j} = \left( B_i^0, B_i^1 \right), \qquad B_i^0 + B_i^1 = 1. \tag{7}$$

Averaging over queries with $n_i = \text{neq} \in \{0, 1\}$,

$$\overline{\mathbf{b}}^{(\text{neq})} = \frac{1}{N_{\text{neq}}} \sum_{i:\, n_i = \text{neq}} \overrightarrow{\mathbf{B}}_i, \tag{8}$$

shows how each class distributes its attention across flexibility types. Queries with $N_{eq} = 0$ devote $\approx 68\%$ of their attention to rigid keys, while queries with $N_{eq} = 1$ devote $\approx 91\%$ of their attention to flexible keys. In short, the model has internalized a modular view of proteins—residues preferentially consult partners that share both structural state and dynamic behavior—even when those partners are distant along the chain.

### 5.3 Conclusion

In this paper, we have demonstrated a novel technique using large ESM protein model embeddings and LSTMs with attention to predict the dynamics of protein sequences and allowing quantitative interpretation. While models like AlphaFold can predict the structure of proteins, to our knowledge there are no models that can predict protein *dynamics*, which is critical for applications such as

Table 4: Attention distribution over SS classes for a query residue of class $s_i \in \{C, H, E\}$. Entries are proportions ($\in [0, 1]$) of attention mass.

| Query SS | $A_i^C$ | $A_i^H$ | $A_i^E$ |
|---|---|---|---|
| C | 0.735 | 0.151 | 0.114 |
| E | 0.558 | 0.215 | 0.227 |
| H | 0.309 | 0.595 | 0.095 |

Table 5: Attention distribution over flexibility classes ($N_{eq}$) conditioned on the query residue's $N_{eq}$ label. Entries are proportions ($\in [0, 1]$).

| Query $N_{eq}$ | $B_i^0$ (to rigid) | $B_i^1$ (to flexible) |
|---|---|---|
| 0 | 0.679 | 0.321 |
| 1 | 0.092 | 0.908 |

drug design. Prior work using deep learning has focused on prediction of static structures, but the movement of amino acids is critical for applications such as drug design. Typically, expensive, time-consuming molecular dynamics simulations are required for predicting flexible protein regions; thus, our approach would help physics and biology researchers by using our deep learning model instead of having to run expensive MD simulations. In future work, we will explore multi-class classification and regression for predicting finer-grained $N_{eq}$ values, which is more useful in practice than binary predictions.

## 6  ETHICS STATEMENT

We acknowledge the use of large language models (LLMs), including OpenAI's GPT-based tools, for editorial assistance (clarification of prose and grammar) and limited code assistance (boilerplate generation, refactoring and debugging suggestions). All conceptual contributions, methodological decisions and experiments were made by the authors. We assume full responsibility for the content, and we validated all outputs.

## 7  REPRODUCIBILITY STATEMENT

We release an anonymous artifact containing fixed train/validation/test chain lists and the test FASTA used in all reported results; and the complete code for training, inference, attention analysis, and benchmarking. All materials are hosted on OpenReview as anonymous supplementary materials. [1]

---

[1]https://anonymous.4open.science/r/ESMfluc-5F08/

# 8 APPENDIX

## 8.1 BASELINE MODELS

A set of classical models was chosen to represent a progression in complexity and are suitable for binary classification and sequence-based tasks as baseline models.

- **Logistic Regression (LR) classifier**: A simple model for binary classification. It provides a strong foundation for determining whether the relationship between features and target labels is linearly separable. Based on the result, we can determine whether more complex, non-linear models are required to achieve better results.

- **Random Forest (RF) classifier**: An ensemble learning classifier composed of multiple decision trees Parmar et al. (2019). By averaging the results from different decision trees, the model is resistant to noise and outliers and able to capture the complex interactions between features.

- **Conditional Random Field (CRF)**: A conditional random field was selected to address the sequential nature of the data specifically. Unlike the previous models, which predict the flexibility of each amino acid independently, a CRF models the conditional probability of a label sequence given the input sequence. This allows it to capture the influence of neighboring amino acids on a given residue's flexibility.

For LR and RF, we engineered a vector of features that was concatenated of:

- Amino Acid Identity: one-hot encoding of the current, previous, and next amino acids.

- Positional Information: normalized position of the amino acid within the sequence ($i/L$, where $i$ is the amino acid index and $L$ is the sequence length)

- Amino Acid Characteristics: numerical encodings for charges (neutral=0, positive=1, negative=-1), polar (polar=1, non-polar=0), and hydrophobic (hydrophobic=1, non-hydrophobic=0).

We also tested the final, high-dimensional embedding for each amino acid from the HuggingFace ESM2's last transformer layers to serve as the feature vector for LR and RF.

For the CRF model, the feature set included the current amino acid, the previous and next amino acids, and a window size of 2 for contextual amino acids.

## 8.2 BATCH SIZE

As shown in Table 6, we investigated the effect of batch size on the performance and efficiency of our ESMFLUC model using ESM2 embeddings with 12 layers and 35 million parameters. Increasing the batch size from 1 to 32 leads to a significant reduction in training time, with the minimum observed at a batch size of 32. However, once the batch size exceeds 32, training time increases. This decline can be explained by exceeding GPU's memory capacity, as indicated by the increase in maximum memory allocated. Memory usage nearly doubles with each double batch size. In terms of model performance, the highest performance was achieved with a batch size of 1. However, when the batch size increases beyond 8, the macro F1 score decreases. This indicates that larger batches converge to a sharp or local minimum, which performs well on the training data, but fails to generalize to the validation set.

## 8.3 LOSS FUNCTION

Additionally, we examined the effect of different loss functions. In addition to standard cross-entropy (CE) and focal loss, we experimented with a neural collapse (NC)-inspired loss, based on a phenomenon observed in highly accurate deep neural networks for classification tasks, where hidden representations in the last layer collapse to their class-specific mean embeddings Papyan et al. (2020). In recent work, an NC-inspired loss function enforced neural collapse during training on a long-tail imbalanced data problem similar to ours, where one term (i.e., NC1) encouraged ESM embeddings from the same function to cluster tightly around a class mean (i.e., intra-class

Table 6: The performance for different batch sizes using FairESM ESM2 with 12 layers.

| Batch Size | Time (s) | Max Memory Allocated (GB) | Macro F1 |
|---|---|---|---|
| 1 | 528 | 0.74 | 78 |
| 2 | 619 | 0.85 | 77 |
| 4 | 270 | 1.17 | 77 |
| 8 | 211 | 1.85 | 76 |
| 16 | 91 | 3.23 | 54 |
| 32 | 77 | 5.97 | 42 |
| 64 | 101 | 11.57 | 35 |
| 128 | 191 | 19.19 | 36 |

compactness) and the other (i.e., NC2) encouraged inter-class separation Luo & Luo (2025). Let $\mathcal{I} = \{i \mid y_i \neq -1\}$ be the set of non-padded tokens, $h_i \in \mathbb{R}^D$ the per-token feature (pre-FC), $y_i \in \{0, \dots, K-1\}$ the class, and $\mu_k$ a learnable class mean. Define cosine similarity and distance:

$$\text{cos\_sim}(a, b) = \frac{a^\top b}{\|a\| \, \|b\|}, \qquad \delta(a, b) = 1 - \text{cos\_sim}(a, b). \tag{9}$$

Let $N_k = \big|\{i \in \mathcal{I} : y_i = k\}\big|$, $w_k = \frac{1}{\sqrt{\max(N_k, 1)}}$. The NC1 term encourages intra-class compactness with $\sqrt{N_k}$ scaling:

$$\mathcal{L}_{\text{NC1}} = \frac{1}{K} \sum_{k=0}^{K-1} \frac{w_k}{\sqrt{\max(N_k, 1)}} \sum_{i \in \mathcal{I}: \, y_i = k} \delta(h_i, \mu_k). \tag{10}$$

NC2 promotes inter-class separation by maximizing the minimum angle between class means. Let $\bar{\mu} = \frac{1}{K} \sum_k \mu_k$, $\tilde{\mu}_k = \mu_k - \bar{\mu}$, and $\hat{\mu}_k = \tilde{\mu}_k / \|\tilde{\mu}_k\|$:

$$c_{kl} = \hat{\mu}_k^\top \hat{\mu}_l, \quad \theta_k = \arccos\Big( \max_{l \neq k} c_{kl} \Big), \qquad \mathcal{L}_{\text{NC2}} = -\frac{1}{K} \sum_{k=0}^{K-1} \theta_k. \tag{11}$$

We combined these with supervised losses on logits ($z_i$) from the classifier head. The total loss is

$$\mathcal{L} = \lambda_{\text{CE}} \mathcal{L}_{\text{sup}} + \lambda_1 \mathcal{L}_{\text{NC1}} + \lambda_2 \mathcal{L}_{\text{NC2}}, \tag{12}$$

with $\mathcal{L}_{\text{sup}} \in \{\mathcal{L}_{\text{CE}}, \mathcal{L}_{\text{FL}}\}$. In the NC-only setting with a centroid head (no logits), we set $\lambda_{\text{CE}} = 0$ and use nearest-centroid inference:

$$\hat{y}_i = \arg \max_k \, \text{cos\_sim}(h_i, \mu_k). \tag{13}$$

Table 7 shows the effect of loss choice on the performance of ESMFLUC. All rows in Table 7 were trained with the ESM backbone *frozen*; only the BiLSTM+attention head (and class means for NC) were updated. Under this constraints, the NC-only (centroid) objective underperforms markedly, which is expected because with frozen features $h_i$ the model cannot reshape the representation geometry; optimizing only the learnable class means $\{\mu_k\}$ provides limited capacity. Supervised losses (CE or focal) achieve strong and very similar performance. If the downstream goal is to recover flexible residues (class 1), the most relevant metric is class-1 recall. Focal loss attains the best class-1 recall (87.6%), followed closely by NC+Focal (87.2%) and CE (86.8%). Adding NC to supervised training (NC+CE or NC+Focal) is roughly on par with the supervised baselines when the backbone is frozen; the supervised term dominates learning. We hypothesize that NC regularization would be more beneficial once the ESM is partially or fully fine-tuned, where NC1/NC2 can actively shape the feature space $h_i$ rather than only the classifier geometry.

Table 7: A comparison of the performance of various loss functions for ESMFLUC.

| Model | Class | Precision | Recall | F1-Score |
|---|---|---|---|---|
| Cross-Entropy | 0 | 80.4 | 72.8 | 76.4 |
| | 1 | 81.1 | 86.8 | 83.8 |
| | macro avg | 80.7 | 79.8 | 80.1 |
| | weighted avg | 80.8 | 80.8 | 80.7 |
| Focal Loss | 0 | 81.1 | 71.6 | 76.0 |
| | 1 | 80.5 | 87.6 | 83.9 |
| | macro avg | 80.8 | 79.6 | 80.0 |
| | weighted avg | 80.8 | 80.7 | 80.5 |
| NC only (centroid) | 0 | 48.8 | 61.2 | 54.3 |
| | 1 | 64.4 | 52.3 | 57.7 |
| | macro avg | 56.6 | 56.7 | 56.0 |
| | weighted avg | 57.8 | 56.1 | 56.3 |
| NC + Focal | 0 | 80.7 | 72.0 | 76.1 |
| | 1 | 80.7 | 87.2 | 83.8 |
| | macro avg | 80.7 | 79.6 | 80.0 |
| | weighted avg | 80.7 | 80.7 | 80.5 |
| NC + CE | 0 | 76.9 | 77.1 | 77.0 |
| | 1 | 82.9 | 82.8 | 82.8 |
| | macro avg | 79.9 | 79.9 | 79.9 |
| | weighted avg | 80.3 | 80.3 | 80.3 |

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
